# MODEL-AGNOSTIC ADVERSARIAL ATTACK AND DEFENSE FOR VISION-LANGUAGE-ACTION MODELS

## ABSTRACT

Vision-Language-Action (VLA) models have achieved revolutionary progress in robot learning, enabling robots to execute complex physical robot tasks from natural language instructions. Despite this progress, their adversarial robustness remains underexplored. In this work, we propose both adversarial patch attack and corresponding defense strategies for VLA models. We first introduce the Embedding Disruption Patch Attack (EDPA), a model-agnostic adversarial attack that generates patches directly placeable within the camera's view. In comparison to prior methods, EDPA can be readily applied to different VLA models without requiring prior knowledge of the model architecture, action space, or the controlled robotic manipulator. EDPA constructs these patches by (i) maximizing the discrepancy of latent representations of adversarial and correspondingly clean visual inputs, and (ii) disrupting the semantic alignment between visual and textual latent representations. Through the optimization of these objectives, EDPA distorts the VLA's interpretation of visual information, causing the model to repeatedly generate incorrect actions and ultimately result in failure to complete the given robotic task. To counter this, we propose an adversarial fine-tuning scheme for the visual encoder, in which the encoder is optimized to produce similar latent representations for both clean and adversarially perturbed visual inputs. Extensive evaluations on the widely recognized LIBERO robotic simulation benchmark demonstrate that EDPA substantially increases the task failure rate of cutting-edge VLA models, while our proposed defense effectively mitigates this degradation.

## 1 INTRODUCTION

Vision-Language-Action (VLA) models (Zitkovich et al., 2023; Team et al., 2024; Kim et al., 2024; Black et al., 2024) built on vision-language foundation models (Touvron et al., 2023; Beyer et al., 2024; Achiam et al., 2023; Liu et al., 2023b) have recently emerged to enable robots to perform complex physical tasks from high-level instructions. Through integrating vision and language understanding, VLAs leverage powerful perceptual and reasoning abilities, allowing robots to generalize to previously unseen environments (Zitkovich et al., 2023).

As the increasing attention on VLA models, concerns about their reliability become particularly urgent, since failures in VLAs deployed on physical robotic platforms can lead to tangible consequences such as robots mishandling objects and resulting in property damage or performing incorrect actions that endanger human safety. Adversarial robustness (Carlini et al., 2019; Goodfellow et al., 2014; Madry et al., 2017) has long been recognized as a critical security challenge in computer vision. Given the VLA models reliance on visual inputs captured from camera, these models are likely to inherit similar vulnerabilities. While adversarial robustness has been extensively investigated in traditional deep learning models, its implications for VLA models remain largely underexplored.

A recent study (Wang et al., 2024) highlighted the vulnerability of VLA models to adversarial attacks. The authors proposed several adversarial patch methods, each employing loss functions tailored to the robotic arm's action trajectory for the OpenVLA (Kim et al., 2024) model controlling a 7-degree-of-freedom (DoF) robotic arm (Zitkovich et al., 2023). Their experiments showed that OpenVLA exhibited almost no resistance to such attacks. However, these attacks depend on stringent requirements: the attacker must have prior knowledge of the victim model's action space, and

full access to all model parameters to compute gradients for generating adversarial patches. These constraints substantially limit the practicality of the attacks in real-world scenarios (see Section 2.2).

To address this limitation, we propose the Embedding Disruption Patch Attack (EDPA), designed to generate adversarial patches that disrupt a VLA's interpretation of visual information. In contrast to prior attacks, EDPA requires only access to the VLA's encoder parameters and does not rely on knowledge of the VLA's architecture or the controlled robot platform. EDPA optimizes patches with two complementary objectives: (i) maximizing the deviation between the latent representations of adversarial and corresponding clean visual inputs, and (ii) disrupting the semantic alignment between the visual latent representation and the corresponding instruction's language latent representation. Through jointly optimizing these objectives, EDPA produces adversarial patches that markedly distort visual understanding in VLAs, leading to a substantial reduction in the success rate of robotic tasks across latest VLA models.

In addition, we introduce a complementary adversarial fine-tuning scheme for the visual encoder to enhance the robustness of VLA models against such attacks. Specifically, all adversarial patches generated during the EDPA optimization process are applied to construct adversarial visual inputs, which are then used to fine-tune the visual encoder. This method encourages the encoder to produce latent representations for adversarial visual inputs that match those of the corresponding clean inputs, while simultaneously ensuring that the fine-tuned encoder preserves performance for clean inputs by maintaining latent representations similar to those produced by the original encoder. Due to our experimental results showed that OpenVLA exhibited the weakest robustness against EDPA, it was chosen as the primary model for defense evaluation. The results demonstrate that adversarial fine-tuning not only strengthens OpenVLA's resistance to EDPA but also significantly improves its robustness against previously proposed untargeted adversarial patch attacks (Wang et al., 2024).

## 2 RELATED WORK

### 2.1 VLA FOR EMBODIMENTS

The concept of embodied AI was introduced by Machinery (1950) to examine whether agents can demonstrate intelligence through interaction with and navigation in complex physical environments, rather than just solving abstract problems in the digital world. In the early stages of developing generalist robots, prevailing approaches (Silva et al., 2021; Nair et al., 2022; Jang et al., 2022; Brohan et al., 2023) primarily relied on reinforcement learning or traditional imitation learning paradigms to acquire task-specific policies.

In recent years, the advent of Large-scale Vision-Language Models (LVLMs) has shifted the paradigm toward leveraging these models to enhance generalization and language grounding in embodied agents, enabling robots to execute tasks directly from natural language instructions. Generalist robotic models such as RT-2 (Zitkovich et al., 2023), Octo (Team et al., 2024), OpenVLA (Kim et al., 2024), and $\pi_0$ (Black et al., 2024), typically built upon LVLMs and commonly referred to as Vision-Language-Action (VLA) models, demonstrate strong generalization across diverse scenes and tasks, facilitating effective transfer to previously unseen scenarios.

### 2.2 ADVERSARIAL ROBUSTNESS IN VLA

Adversarial robustness is a fundamental challenge in securing deep learning models in computer vision. It concerns the resilience of these models to malicious input, known as adversarial attacks. Although primitive adversarial attacks (Goodfellow et al., 2014; Madry et al., 2017; Croce & Hein, 2020; Carlini & Wagner, 2017) demonstrate strong effectiveness in deceiving models through imperceptible pixel-level additive noise, they are often impractical in physical-world applications, as adversaries typically have limited access to the resources (Sharma et al., 2022) (e.g., adversary may not be able to directly modify the pixel of image). In this context, adversarial patch attacks (Brown et al., 2017; Karmon et al., 2018; Liu et al., 2018; Li et al., 2019) focus on manipulating a contiguous region of an image with perceptible but implementable perturbations (e.g., printed as stickers). Given that VLA models incorporate visual input, their adversarial robustness raises concerns analogous to those observed in other computer vision tasks, but studies in this field remain limited.

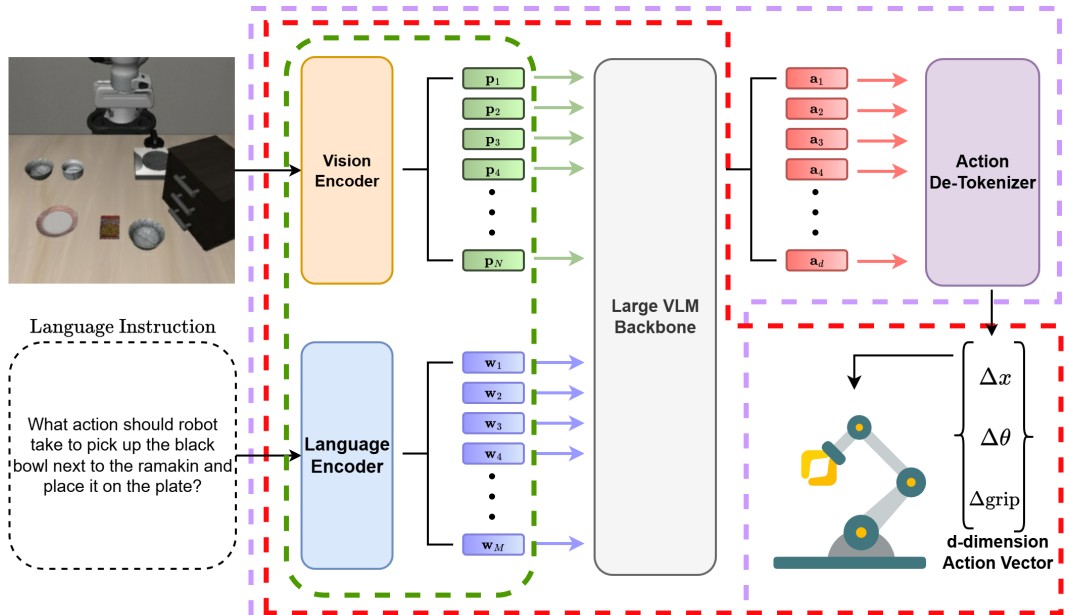

Figure 1: **Overview of VLA architecture and patch attack requirements.** Given a visual observation and a language instruction, the VLA model first encodes the inputs into token-level latent representations. These representations are processed by the LVLM to produce action tokens, which are subsequently decoded into executable actions for the robotic platform. The colored dashed lines highlight the prior knowledge and/or access required by different patch attacks for various modules within the VLA: green for EDPA, purple for UADA, and red for UPA.

To the best of our knowledge, Wang et al. (2024) conducted the first systematic study on the robustness of VLA models against adversarial patch attacks. They proposed two untargeted adversarial patch attacks to explore the adversarial robustness of VLA: the Untargeted Action Discrepancy Attack (UADA) and the Untargeted Position-aware Attack (UPA). These attacks generate adversarial patches that aim to cause OpenVLA to produce action trajectories that deviate from the intended trajectories when controlling a 7-DoF robotic arm. UADA exploits OpenVLA's action space, where differences in action token values are correlated with differences in the magnitudes of the decoded actions in physical space, forcing OpenVLA to output tokens whose numeric values deviate maximally from the true action token. In contrast to manipulating the numeric values of action tokens, UPA directly operates on the first three components of the action vector controlling a 7-DoF robotic arm. These components represent the 3D directional movements of the robotic arm, and UPA forces OpenVLA to produce action vectors that deviate from the intended ones along these dimensions.

| Category | Requirement | UADA (Wang et al., 2024) | UPA (Wang et al., 2024) | **EDPA (Ours)** |
|---|---|---|---|---|
| Knowledge | Action Space | ✓ | ✗ | ✗ |
| | Robotic Manipulator | ✗ | ✓ | ✗ |
| Access | Encoder Parameters | ✓ | ✓ | ✓ |
| | LVLM Parameters | ✓ | ✓ | ✗ |

Table 1: Comparison of attack requirements between UADA, UPA, and EDPA.

These attacks are constrained by stringent assumptions that hinder their applicability to other VLA models and robotic manipulators. UADA requires precise knowledge of the victim model's action space and assumes that differences in action token values correspond to differences in decoded action magnitudes. UPA requires knowledge of the robotic manipulator and the structure of its action vector to locate the components corresponding to 3D directional movements of the robotic arm. They also require access to all the parameters VLA to compute gradients. In comparison, our

proposed EDPA can operate without detailed knowledge of the victim model's action space or the controlled robotic manipulator, relying solely on access to the encoder parameters, which makes it more practical for real-world scenarios (see Table 1 and Figure 1).

## 3 METHODOLOGY

### 3.1 PRELIMINARIES

**Vision-language Action Models.** The architecture of SOTA VLA models built on top of large-scaled LVLM commonly consist of three main components:

(1) A **visual encoder** $\mathcal{E}_v(\cdot)$ that transforms the visual input $v$ (i.e., an image) captured by a camera into a sequence of *image patch embeddings* and projects them into the input space of the language model:

$$\mathcal{E}_v(v) = [\, \mathbf{p}_1, \mathbf{p}_2, \ldots, \mathbf{p}_N \,], \quad \mathbf{p}_i \in \mathbb{R}^d,$$

where $\mathbf{p}_i$ denotes the $d$-dimensional embedding of the $i$-th image patch, and $N$ is the total number of patches.

(2) A **language encoder** $\mathcal{E}_t(\cdot)$ that tokenizes the natural language instruction $t$ into a sequence of textual tokens and encodes them into *language tokens embeddings*:

$$\mathcal{E}_t(t) = [\, \mathbf{w}_1, \mathbf{w}_2, \ldots, \mathbf{w}_M \,], \quad \mathbf{w}_j \in \mathbb{R}^d,$$

where $\mathbf{w}_j$ denotes the $d$-dimensional embedding of the $j$-th token, and $M$ is the total number of tokens in the instruction.

(3) The **LVLM backbone** $f(\cdot)$ processes the concatenated *image patch embeddings* and *language token embeddings* to produce a sequence of action tokens:

$$A = f\big(\mathcal{E}_v(v), \mathcal{E}_t(t)\big)$$

where $A$ denotes the generated action tokens sequence, which can be further converted into an executable action vector for the robot manipulator through a detokenizer.

**Adversarial Patch Attack.** An adversarial patch attack is a specific type of adversarial attack in which the perturbation is applied to a localized area of the image, known as an adversarial patch. The patch is typically of a fixed shape (e.g., square or arbitrary) and can be randomly placed at any location within the image to mislead the model.

Formally, given a clean visual input $v \in [0,1]^{H \times W \times C}$ and an adversarial patch $\delta \in [0,1]^{h \times w \times C}$, the adversarial patch attack applies the patch $\delta$ to $v$ by replacing the pixel values within the patch region:

$$v \oplus \delta = (1 - p) \odot v + p \odot \delta, \tag{1}$$

where $p$ is a binary mask indicating the patch's shape and location, and $h$ and $w$ are the height and width of the patch. The operator $\odot$ denotes Hadamard element-wise multiplication.

### 3.2 EMBEDDING DISRUPTION PATCH ATTACK

A number of studies (Zhang et al., 2022; Zhao et al., 2023; Bagdasaryan et al., 2024; Lu et al., 2023; Zhang et al., 2025) have shown that adversarial attacks targeting embedding representations are generally highly effective against LVLMs. Since VLAs are built upon LVLMs, they likely inherit similar vulnerabilities. As these attacks typically rely on traditional pixel-level perturbations, directly applying them to VLAs that interact with the physical environment is impractical. Motivated by this insight, we propose an untargeted adversarial patch method, **Embedding Disruption Patch Attack (EDPA)**, which specifically targets latent representations within VLA models. This method requires no access to the VLM backbone or prior knowledge of the model architecture and action space and is agnostic to the type of robotic manipulator. The learning objective of the EDPA comprises two components: (1) maximizing the discrepancy between the *image patch embeddings* of the clean visual input $v$ and the adversarial visual input $v'$, and (2) disrupting the original semantic alignment between the *image patch embeddings* of $v$ and the *language token embeddings* of

the language instruction $t$. To this end, we introduce the corresponding loss functions: the patch contrastive loss and the image-instruction alignment loss.

**Patch Contrastive Loss.** The patch contrastive loss is designed to quantify the discrepancy between the image patch embeddings of the clean and perturbed visual inputs. Specifically, given a clean input visual input $v$ and its perturbed counterpart $v' = v \oplus \delta$, we measure the embedding deviation between $\mathcal{E}_v(v)$ and $\mathcal{E}_v(v')$. Inspired by the InfoNCE (Oord et al., 2018), we define the loss function as follows:

$$\mathcal{L}_{\text{patch}}(\mathcal{E}_v(v), \mathcal{E}_v(v')) = -\frac{1}{N} \sum_{i=1}^{N} \log \frac{\exp\left(\cos(\mathbf{p}_i, \mathbf{p}'_i)/\tau\right)}{\sum_{j=1}^{N} \exp\left(\cos(\mathbf{p}_i, \mathbf{p}'_j)/\tau\right)}, \tag{2}$$

where $\mathbf{p}_i$ and $\mathbf{p}'_i$ denote the $i$-th *image patch embeddings* in $\mathcal{E}_v(v)$ and $\mathcal{E}_v(v')$, respectively. Here, $\tau$ is a scalar hyperparameter, and $\cos(\cdot, \cdot)$ is the cosine similarity function.

**Image-Instruction Alignment Loss.** In contrast to the patch contrastive loss that focuses on the discrepancy of the *image patch embeddings*, the image-instruction alignment loss measures the change in semantic alignment between the *image patch embeddings* and the *language token embeddings* of the corresponding language instruction. Formally, given a language instruction $t$ corresponding to the visual input $v$, we measure the change in alignment between the *image patch embeddings* $\mathcal{E}_v(v)$ and $\mathcal{E}_v(v')$ with respect to the *language token embeddings* $\mathcal{E}_t(t)$. To this end, we define the loss function as follow:

$$\mathcal{L}_{\text{align}}(\mathcal{E}_v(v), \mathcal{E}_v(v'), \mathcal{E}_t(t)) = \frac{1}{N \times M} \sum_{i=1}^{N} \sum_{j=1}^{M} \left| \cos\left(\mathbf{p}_i, \mathbf{w}_j\right) - \cos\left(\mathbf{p}'_i, \mathbf{w}_j\right) \right|, \tag{3}$$

where $\mathbf{w}_j$ denotes the $j$-th *language token embedding* in $\mathcal{E}_t(t)$.

**Adversarial Patch Generation.** To construct a universal adversarial patch $\delta$, we formulate an optimization problem that jointly maximizes the patch contrastive loss (equation 2) and the image-instruction alignment loss (equation 3) as following:

$$\delta^* = \arg\max_{\delta} \mathbb{E}_{v \sim \mathcal{D}} \left[ \alpha_1 \cdot \mathcal{L}_{\text{patch}}(\mathcal{E}_v(v), \mathcal{E}_v(v \oplus \delta)) + (1 - \alpha_1) \cdot \mathcal{L}_{\text{align}}(\mathcal{E}_v(v), \mathcal{E}_v(v \oplus \delta), \mathcal{E}_t(t)) \right], \tag{4}$$

where $\mathcal{D}$ is the data distribution of visual inputs and $\alpha_1 \in [0, 1]$ is a hyperparameter controlling the relative contributions of each loss function. In practice, we find this objective effective in constructing adversarial patches that degrade the ultimate performance of the VLA model.

### 3.3 ADVERSARIAL FINETUNING ON VISUAL ENCODER

In the Section 3.2, we demonstrate that an effective adversarial patch $\delta$ can be derived by directly targeting the embedding space of the VLA. In this section, we present a complementary adversarial finetuning scheme aimed at improving the robustness of the visual encoder $\mathcal{E}_v(\cdot)$ within the VLA.

The finetuning scheme incorporates adversarial visual inputs constructed from adversarial patches $\delta$ generated by EDPA. Instead of relying solely on the final optimized $\delta$, the training process utilizes all intermediate patches produced during the EDPA training. Throughout the process, the current $\delta$ is applied to the visual inputs to generate perturbed samples (refer to equation 1), which are then used to optimize the visual encoder. In addition, we adopt a fixed reset frequency for $\delta$, periodically reinitializing the patch during training to prevent overfitting to a specific patch and to ensure that the visual encoder is exposed to a diverse set of adversarial patches.

The learning objective of the finetuning process mainly consists of two complementary objectives: (i) to encourage the fine-tuned visual encoder to produce latent representations of adversarially perturbed visual inputs that are close to those of the corresponding clean visual inputs produced by the original visual encoder, thereby enhancing the visual encoder's robustness against adversarial patches; and (ii) to ensure that the latent representations generated by the fine-tuned visual encoder

---

**Algorithm 1:** Adversarial Finetuning on Visual Encoder

---

**Input:** Original visual encoder $\mathcal{E}_v^{\text{orig}}$, language encoder $\mathcal{E}_t$, robotic dataset $\mathcal{D}$, hyperparameter
$\qquad \alpha_1$, hyperparameter $\alpha_2$, step size $\eta_\delta$, patch reset frequency $\varphi$, inner attack iterations $K$,
$\qquad$ learning rate $\eta$, max training iterations $T$
**Output:** visual encoder $\mathcal{E}_v^*$
Initialize $\mathcal{E}_v \leftarrow \mathcal{E}_v^{\text{orig}}$, $\delta \sim \mathcal{U}(0,1)$;
**for** $i = 1$ **to** $T$ **do**
$\quad$ Sample minibatch $(v, t) \subset \mathcal{D}$;
$\quad$ **if** $i \bmod \varphi = 0$ **then**
$\quad\quad$ Reinitialize $\delta \sim \mathcal{U}(0,1)$;
$\quad$ **for** $k = 1$ **to** $K$ **do**
$\quad\quad$ $\mathcal{J} \leftarrow \alpha_1 \cdot \mathcal{L}_{\text{patch}}(\mathcal{E}_v(v), \mathcal{E}_v(v \oplus \delta)) + (1 - \alpha_1) \cdot \mathcal{L}_{\text{align}}(\mathcal{E}_v(v), \mathcal{E}_v(v \oplus \delta), \mathcal{E}_t(t))$;
$\quad\quad$ $\delta \leftarrow \text{clip}(\delta + \eta_\delta \cdot \text{sign}(\nabla_\delta \mathcal{J}), 0, 1)$
$\quad$ $\mathcal{L} \leftarrow \alpha_2 \cdot \|\mathcal{E}_v(v) - \mathcal{E}_v^{\text{orig}}(v)\|_2^2 + (1 - \alpha_2) \cdot \|\mathcal{E}_v(v \oplus \delta) - \mathcal{E}_v^{\text{orig}}(v)\|_2^2$;
$\quad$ Update $\mathcal{E}_v$ by gradient descent with learning rate $\eta$;
**return** $\mathcal{E}_v$;

---

on clean visual inputs remain consistent with those generated by the original visual encoder, thereby preserving fidelity on clean visual inputs. As a result, the fine-tuned visual encoder can be directly integrated into the VLA without any modification or further fine-tuning of the LVLM backbone, mitigating the impact of adversarial patches while preserving overall performance.

To formalize the learning objective of the finetuning scheme, let $\mathcal{E}_v^{\text{orig}}(\cdot)$ denote the original visual encoder without being adversarial finetuned. The objective is to optimize the parameters of the updated visual encoder by solving the following optimization problem:

$$\mathcal{E}_v^* = \arg\min_{\mathcal{E}_v} \mathbb{E}_{v \sim \mathcal{D}} \left[ \alpha_2 \cdot \left\| \mathcal{E}_v(v) - \mathcal{E}_v^{\text{orig}}(v) \right\|_2^2 + (1 - \alpha_2) \cdot \left\| \mathcal{E}_v(v \oplus \delta) - \mathcal{E}_v^{\text{orig}}(v) \right\|_2^2 \right] \quad (5)$$

where $\delta$ denotes an adversarial patch generated by EDPA during its training procedure, and $\alpha_2 \in [0, 1]$ controls the relative contributions of the two learning objectives. The pseudocode of our fine-tuning scheme is shown in Algorithm 1.

## 4 EXPERIMENT

### 4.1 EXPERIMENT SETTINGS

**Dataset.** The adversarial patch generation through EDPA are conduct on LIBERO (Liu et al., 2023a), a simulation dataset specifically designed for robotic manipulation. The datasets comprises four distinct task suites: Spatial, Object, Goal, and Long.

**Victim Models.** We evaluate recent open-source VLA models, including OpenVLA Kim et al. (2024), OpenVLA-OFT (Kim et al., 2025), and $\pi_0$ (Black et al., 2024), all of which provide fine-tuned variants for LIBERO. Specifically, OpenVLA and OpenVLA-OFT each offer separate fine-tuned models for individual task suites, whereas $\pi_0$ provides a single model fine-tuned across all task suites.

**Baseline Method.** Given the limited research in this domain, no existing baseline directly matches our experimental setting. The most relevant untargeted adversarial patch attacks are UADA and UPA. However, these attacks are difficult to transfer to models other than OpenVLA due to their stringent application requirements. Therefore, we compare EDPA with UADA and UPA in terms of attack performance only on the OpenVLA model, while also evaluating the effectiveness of our defense method against them. Details of both experiments are provided in Section 4.2. For general experiments, we use a random noise baseline following Wang et al. (2024), where patches are sampled from a Gaussian distribution $\mathcal{N}(0, 1)$ and evaluated under the same settings as EDPA.

**Hyperparameter Settings.** In all experiments, the sizes of both adversarial and noise patches are fixed at 50×50 pixels, following Wang et al. (2024), while VLAs commonly receive visual inputs at a resolution of 224×224. During the generation of adversarial patches with EDPA, the hyperparameter $\alpha_1$, which controls the relative contribution of each loss, is set to 0.8, and the number of inner attack iterations $K$ is fixed at 1. Since the two losses may operate on different scales, exponential moving average (EMA) normalization is applied to each loss to ensure that $\alpha_1$ accurately governs their relative contributions. The step size $\eta_\delta$ is set to $2/255$, and training proceeds for a maximum of $T = 50{,}000$ iterations with a batch size of 16. For the adversarial fine-tuning scheme, we set $\alpha_2 = 0.5$ to balance the two learning objectives, and the patch reset frequency $\varphi$ is set to 1000. The visual encoder is optimized using the Adam optimizer with a learning rate of $\eta = 1 \times 10^{-5}$. The sensitivity to some of these hyperparameter settings are reported in Appendix C.

**Simulation and Evaluation Metric.** We evaluate our methods on all four task suites of the LIBERO simulation benchmark, with each suite comprising 10 tasks and 50 executions per task. Performance is measured using the Failure Rate (FR) metric following Wang et al. (2024), representing the proportion of tasks that were not successfully completed after a certain number of steps (defined as Failure Rate = 1 - Success Rate). To account for stochastic variability, reported FRs are averaged over three experiments with different random seeds, following Kim et al. (2024).

## 4.2 EVALUATING ON SINGLE-CAMERA VLA

In this subsection, we focus on evaluating the performance of our attack and defense methods on a single-camera VLA. We adopt OpenVLA as the representative model, which relies solely on visual input from the primary camera. In our evaluation, we measure the performance of OpenVLA before and after adversarial fine-tuning of the visual encoder when subjected to various patch-based attacks in the libero simulation benchmark.

As shown in Table 2, the OpenVLA without adversarial fine-tuned visual encoder demonstrates almost no resilience to the adversarial patch attacks designed to mislead the model. Compared to common baselines, the adversarial patches generated via EDPA causes OpenVLA to increase its average failure rate on LIBERO tasks by approximately around 74.7% relative to the clean condition and by 53.0% relative to the random noise patch. For comparison with untargeted adversarial patch attacks specifically designed for OpenVLA, we also evaluated adversarial patches generated by UADA and UPA on the LIBERO dataset. The results show that UADA, UPA, and EDPA differ only marginally in effectiveness, as OpenVLA demonstrates minimal resilience against such adversarial patches.

We then evaluate OpenVLA models integrated with an adversarially fine-tuned visual encoder against patch attacks. The results in Table 2 demonstrate such models exhibit substantially reduced failure rates, with average decreases of 34.2% against EDPA and 21.5% against random noise patches in the LIBERO simulation environment. In parallel, the results demonstrate that adversarially fine-tuning the visual encoder not only mitigates the impact of EDPA but also confers improved robustness to OpenVLA against adversarial patches produced by other methods, with reductions in failure rates of 19.1% for UADA and 36.0% for UPA. Importantly, this improvement results in only a minor 1.6% increase in failure rate under clean conditions, reflecting the well-known trade-off between robustness and standard performance.

## 4.3 EVALUATING ON MULTI-CAMERA VLA

In this subsection, we evaluate the effectiveness of EDPA attacks on multi-camera VLAs. We use OpenVLA-OFT and $\pi_0$ as the victim models, both of which rely on visual inputs from the primary camera as well as the wrist camera. In contrast to the primary camera, the wrist camera's viewpoint changes substantially as the robot arm moves. Since real-time alignment of primary and wrist camera observations for the same patch is not feasible, we apply separate adversarial patches to each camera independently for evaluation. As UADA and UPA are difficult to transfer to models other than OpenVLA due to their stringent application requirements, we do not include them in the comparisons in this subsection.

As shown in Table 3, EDPA increases the average failure rate on LIBERO tasks by approximately 62.0% for OpenVLA-OFT and 31.4% for $\pi_0$. Compared with random noise, EDPA induces markedly higher increases in average failure rates of around 50.5%, and 26.5% for OpenVLA-OFT,

Table 2: **Attack and defense performance on OpenVLA.** Average failure rates (FR) of OpenVLA models across four task suites in the LIBERO benchmark under different attacks, reported before and after adversarial fine-tuning.

| Source | Method | Failure Rate (FR ↑) | |
|--------|--------|---------------------|---|
| | | Original | Adversarial Finetuned |
| **Spatial** | Clean | $14.1 \pm 0.5$ | $17.9 \pm 0.8$ |
| | Random | $34.8 \pm 1.1$ | $19.4 \pm 1.4$ |
| | UADA | $98.9 \pm 0.1$ | $\textbf{65.4} \pm \textbf{1.0}$ |
| | UPA | $99.1 \pm 0.3$ | $46.6 \pm 1.0$ |
| | EDPA (Ours) | $\textbf{100.0} \pm \textbf{0.0}$ | $39.4 \pm 1.0$ |
| **Object** | Clean | $12.0 \pm 0.4$ | $17.3 \pm 0.7$ |
| | Random | $39.2 \pm 1.4$ | $16.0 \pm 0.9$ |
| | UADA | $92.5 \pm 0.7$ | $\textbf{58.8} \pm \textbf{1.4}$ |
| | UPA | $92.1 \pm 0.8$ | $43.9 \pm 1.4$ |
| | EDPA (Ours) | $\textbf{100.0} \pm \textbf{0.0}$ | $58.6 \pm 0.6$ |
| **Goal** | Clean | $26.9 \pm 1.5$ | $22.8 \pm 0.4$ |
| | Random | $37.9 \pm 0.7$ | $23.0 \pm 1.1$ |
| | UADA | $98.6 \pm 0.1$ | $\textbf{91.6} \pm \textbf{0.4}$ |
| | UPA | $98.9 \pm 0.2$ | $68.3 \pm 1.7$ |
| | EDPA (Ours) | $\textbf{100.0} \pm \textbf{0.0}$ | $73.9 \pm 1.1$ |
| **Long** | Clean | $48.1 \pm 1.9$ | $49.0 \pm 0.3$ |
| | Random | $74.9 \pm 2.4$ | $50.2 \pm 0.5$ |
| | UADA | $99.6 \pm 0.2$ | $\textbf{97.4} \pm \textbf{0.4}$ |
| | UPA | $99.6 \pm 0.3$ | $86.7 \pm 0.9$ |
| | EDPA (Ours) | $\textbf{100.0} \pm \textbf{0.0}$ | $91.2 \pm 0.5$ |

Table 3: **Attack performance on Other VLAs.** The average failure rates (FR) of various fine-tuned VLA models on the four task suites in the LIBERO simulation benchmark under different perturbation levels.

| Source | Method | Failure Rate (FR) ↑ | |
|--------|--------|---------------------|---|
| | | OpenVLA-OFT | $\pi_0$ |
| **Spatial** | Clean | $1.4 \pm 0.4$ | $3.5 \pm 0.3$ |
| | Random | $8.1 \pm 2.1$ | $4.0 \pm 0.9$ |
| | EDPA | $\textbf{39.7} \pm \textbf{0.9}$ | $\textbf{29.8} \pm \textbf{1.6}$ |
| **Object** | Clean | $2.0 \pm 0.0$ | $2.3 \pm 0.5$ |
| | Random | $15.9 \pm 0.4$ | $4.3 \pm 0.3$ |
| | EDPA | $\textbf{52.3} \pm \textbf{0.8}$ | $\textbf{39.5} \pm \textbf{1.7}$ |
| **Goal** | Clean | $2.8 \pm 0.7$ | $12.0 \pm 1.6$ |
| | Random | $5.1 \pm 0.1$ | $17.5 \pm 1.3$ |
| | EDPA | $\textbf{80.8} \pm \textbf{0.4}$ | $\textbf{44.3} \pm \textbf{2.0}$ |
| **Long** | Clean | $4.9 \pm 0.6$ | $40.8 \pm 1.6$ |
| | Random | $28.1 \pm 2.4$ | $51.9 \pm 0.8$ |
| | EDPA | $\textbf{86.4} \pm \textbf{1.9}$ | $\textbf{70.7} \pm \textbf{1.6}$ |

and $\pi_0$, respectively. These results indicate that EDPA remains highly effective against other (SOTA) multi-camera VLA models. Although the differences between OpenVLA, OpenVLA-OFT, and $\pi_0$ extend beyond camera settings, the results suggest that VLAs processing multiple camera views may exhibit improved robustness to adversarial patches, potentially due to the additional visual information provided by multiple viewpoints.

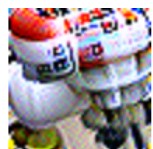 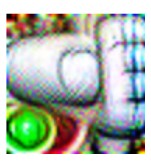 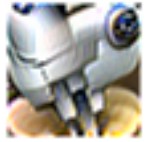  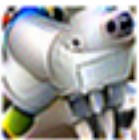 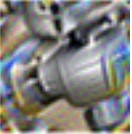

(a) EDPA patches on OpenVLA, OpenVLA-OFT, and $\pi_0$.

(b) Patches generated through UADA and UPA on OpenVLA.

Figure 2: **Visualization of Patch.** All the patches are generated on LIBERO dataset.

## 5 PATCH VISUALIZATION AND DISCUSSION

The Figure 2 illustrates representative adversarial patches produced by various patch-based attack methods on the LIBERO dataset. Specifically, Figure 2a presents patches generated by EDPA on OpenVLA, OpenVLA-OFT, and $\pi_0$, whereas Figure 2b displays patches produced by UADA and UPA on OpenVLA. An interesting observation from these visualizations is that all the generated patches consistently exhibit structural patterns reminiscent of a robotic arm. In particular, the patches generated by EDPA targeting OpenVLA and OpenVLA-OFT exhibit visual patterns that more closely resemble a robotic manipulator than those targeting $\pi_0$. In light of these observations, we propose a hypothesis slightly different from that suggested by Wang et al. (2024). The robotic dataset used for pretraining and fine-tuning VLA models is considerably smaller in scale compared to general-purpose datasets. In addition, robotic data collection is typically constrained to a limited set of camera viewpoints, with the primary camera position remaining fixed. The limited scale and restricted camera viewpoints result in robotic arms being almost always present in the captured frames, which in turn causes the visual encoder of VLA models to overfit to their appearance.

This hypothesis also helps explain our experimental observations. OpenVLA-OFT and $\pi_0$ show greater robustness to EDPA compared to OpenVLA, likely because OpenVLA, which processes only single-camera information, was trained primarily on images from the primary camera view. However, even though both OpenVLA-OFT and $\pi_0$ can process multiple camera views, $\pi_0$ demonstrates superior robustness. This is likely because OpenVLA-OFT, even with wrist camera data included during fine-tuning, is based on the original OpenVLA model whose visual encoder had already overfitted during pretraining. In contrast, $\pi_0$ incorporates wrist camera data from the pretraining stage, providing greater diversity in the visual encoder's training data and thereby mitigating overfitting.

## 6 LIMITATION

Despite the effectiveness of our methods, there are still some limitations that need to be acknowledged: (i) in multi-camera settings, we cannot compute the alignment of observations from different camera views for the same patch in real time. This limits EDPA's ability to optimize patches under conditions that fully reflect their physical observation, potentially reducing attack effectiveness; and (ii) object position information cannot be directly obtained from static data, meaning the adversarial patch may occasionally occlude important objects. In such cases, our adversarial fine-tuning scheme on the visual encoder could potentially have a negative impact on the encoder's performance.

## 7 CONCLUSION

In this study, we investigated the robustness of VLA models against adversarial patch attacks, a critical yet underexplored threat to their reliability. We first introduced a novel patch generation method, targeting the latent representation space of VLA models. We then proposed an adversarial fine-tuning strategy to enhance VLA robustness against such attacks. Our empirical results reveal significant vulnerabilities in current SOTA models and demonstrate that the proposed defense can effectively mitigate these threats. We hope this work will inspire future research efforts toward developing robust and secure vision-language embodied agents capable of safe deployment in complex real-world environments.

## REPRODUCIBILITY STATEMENT

Following the details provided in the main text, we ensure the reproducibility of our results by providing the loss functions, pseudocode, and the specific hyperparameters used in our experiments. Researchers can replicate our experiments and verify our findings using these descriptions and settings. While the results may not be exactly identical due to randomness factor in the experiments, they are expected to fall within a reasonable range. In addition, we will release our codebase upon acceptance of this paper.

## ETHICS STATEMENT

This work demonstrates a potential security threat in VLAs: carefully crafted adversarial patches can significantly degrade the performance of a VLA when placed within the camera's view, causing the embodied agent to misinterpret visual information in the physical environment. This not only reduces the agent's task success rate but also results in unexpected movement trajectories during task execution, posing hazards such as object mishandling, property damage, or actions endangering human safety. In this work, we do provide a method for generating such adversarial patches. We acknowledge that similar methods could be misused maliciously, but our intention is to highlight potential vulnerabilities in VLAs and to encourage the development of defenses methods against such attacks. Our goal is to promote the creation of embodied AI systems that are more robust, secure, and reliable.

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

# APPENDIX

## A    ATTENTION VISUALIZATION

To gain deeper insight into how EDPA disrupts the performance of VLAs, we examine the average attention weights from linguistic tokens to visual input locations. Specifically, we use OpenVLA as the victim model and visualize the averaged attention weights across all heads in both the first and last layers under three perturbation conditions: clean, random, and EDPA. Here, clean denotes unperturbed inputs, random corresponds to patches generated from Gaussian noise, and EDPA refers to adversarial patches produced by our method (Section 3.2). To isolate their effects, all perturbation patches are fixed at the top-left corner of the visual input.

In Figure 3, we present the first-layer average attentions of OpenVLA for three sampled linguistic tokens with respect to their corresponding image regions. For clean samples, the tokens correctly attend to the robot arm and other salient objects visible in the camera view. Under random perturbations, the attention distributions remain largely stable. In contrast, when adversarial patches generated by EDPA are applied, the first-layer attentions shift dramatically: the tokens concentrate predominantly on the patch location, while their focus on the originally relevant objects and the robot arm is markedly reduced.

Figure 4 shows that this phenomenon also persists in the final attention layer of OpenVLA. Compared to the first layer, the linguistic tokens in the last layer exhibit more localized focus on specific regions within the image. However, when adversarial patches derived from EDPA are introduced, a clear change in the attention distribution emerges: the patch location receives disproportionately high attention, while focus on originally important objects is markedly diminished.

In summary, these visualizations demonstrate that adversarial patches generated by EDPA can substantially distort the attention distribution of linguistic tokens over the visual input, thereby undermining model performance.

## B    TRANSFERABILITY OF EDPA

In practice, an adversary may not have full access to the finetuned model or the dataset of downstream task. Here, we evaluate the transferability of EDPA under two scenarios: (1) *cross-dataset transferability*, where the adversary lacks access to downstream task data; and (2) *cross-model transferability*, where the adversary has access only to the victim's base model.

To evaluate *cross-dataset transferability*, we generate adversarial patches on LIBERO-Spatial and apply them to the other three task suites within the LIBERO simulation benchmark. As shown in Table 4, EDPA demonstrates strong dataset-level transferability, substantially increasing the average failure rates about 74.7%, 52.4%, and 33.7% for OpenVLA, OpenVLA-OFT, and $\pi_0$, respectively. Interestingly, the attack performance of EDPA is comparable to that observed under fully white-box

Figure 3: Average attention weights of each linguistic token to the primary camera input in the first layer of OpenVLA.

Table 4: The average failure rates (FR) of different fine-tuned VLA models on the other three task suites in the LIBERO simulation benchmark on EDPA adversarial patches derived from LIBERO-Spatial.

| Source | Failure Rate (FR) | | |
|---|---|---|---|
| | OpenVLA | OpenVLA-OFT | $\pi_0$ |
| **Object** | $100.0 \pm 0.0$ | $32.2 \pm 1.0$ | $30.3 \pm 1.2$ |
| **Goal** | $100.0 \pm 0.0$ | $71.8 \pm 0.9$ | $42.7 \pm 2.2$ |
| **Long** | $100.0 \pm 0.0$ | $61.4 \pm 1.6$ | $71.9 \pm 0.9$ |

settings, suggesting that EDPA requires minimal knowledge of the training data and that an adversary can effectively compromise VLA performance even without access to data from the targeted task.

In parallel, we examine the *cross-model transferability* of EDPA in a scenario where the adversary has no access to the victim model but can leverage its corresponding base model to generate adversarial patches, which are subsequently transferred to the downstream variant. As summarized in Table 5, these transferred patches increase the average failure rates around 49.8%, 26.98%, and 9.3% for OpenVLA, OpenVLA-OFT, and $\pi_0$, respectively, evaluated across the four task suites within the LIBERO simulation benchmark. These findings indicate that, although the *cross-model transferability* of EDPA is generally lower than that observed under fully white-box settings or in cross-dataset

**Instruction: What action should the robot take to pick up the black bowl next to the ramakin and place it on the plate?**

Figure 4: Average attention weights of each linguistic token to the primary camera input in the last layer of OpenVLA.

Table 5: The average failure rates (FR) of different fine-tuned VLA models on the four task suites in the LIBERO simulation benchmark on EDPA adversarial patches derived from the base model.

| Source | Failure Rate (FR) | | |
|---|---|---|---|
| | OpenVLA | OpenVLA-OFT | $\pi_0$ |
| **Spatial** | $71.5 \pm 1.4$ | $12.3 \pm 0.6$ | $7.1 \pm 0.3$ |
| **Object** | $69.4 \pm 1.7$ | $31.6 \pm 1.6$ | $10.0 \pm 1.4$ |
| **Goal** | $67.4 \pm 1.7$ | $32.8 \pm 0.2$ | $22.7 \pm 0.8$ |
| **Long** | $91.9 \pm 1.8$ | $42.3 \pm 3.2$ | $55.8 \pm 1.1$ |

transfer scenarios, it nonetheless maintains significant attack effectiveness against downstream VLA variants.

## C  ABLATION STUDY

### C.1  IMPACT OF ADVERSARIAL PATCH SIZE ON VLA PERFORMANCE

In our primary experiments, we fix the patch size at $50 \times 50$ pixels, consistent with prior work, to enable a fair comparison with baseline. In this section, we further investigate how varying the size of adversarial patches generated by EDPA influences the performance of VLA models. In

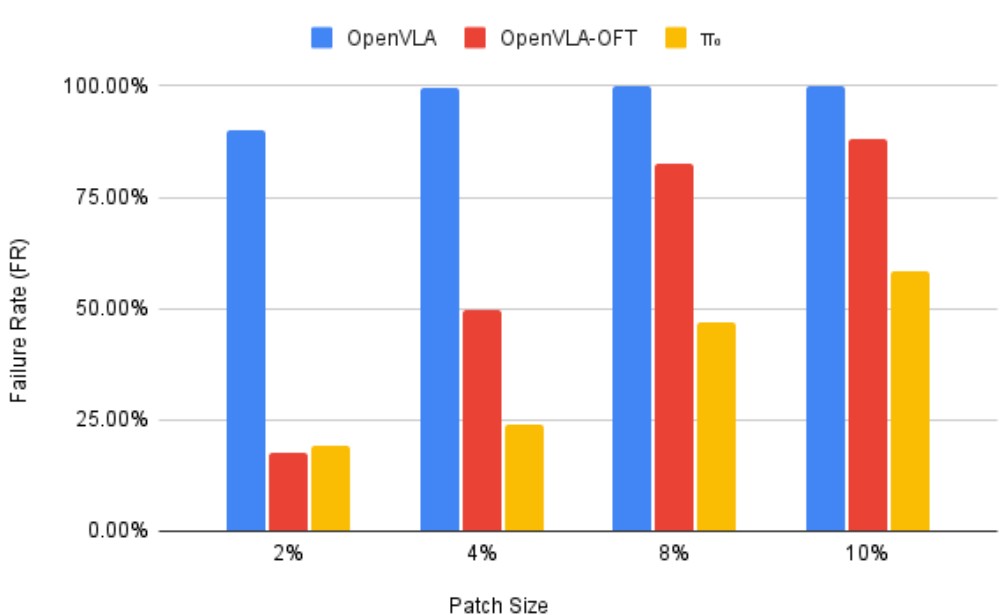

Figure 5: **Impact of patch size.** The figure shows how varying patch sizes of EDPA affect the average failure rate (FR) across different VLA models on the LIBERO simulation benchmark.

our experimental setup, we generate adversarial patches using EDPA with sizes determined as 2%, 4%, 8%, and 10% of the total visual observation area, study their impact on different VLA model performance. We report the average failure rate across four task suites in the LIBERO simulation benchmark for each model under different patch sizes in Figure 5.

The experimental results demonstrate that the performance of all VLA models is consistently affected by the size of adversarial patches. As the patch size increases, the failure rate of all models in executing robotic tasks also rises. This trend suggests that larger adversarial patches lead to stronger degradation of the model's performance.

## C.2 ANALYZING THE IMPACT OF $\alpha_1$ ON EDPA'S EFFECTIVENESS IN VLAS

To investigate how the hyperparameter $\alpha_1$ influences the effectiveness of EDPA, we conduct a series of experiments by systematically varying $\alpha_1$ while keeping all other settings fixed. This hyperparameter controls the relative contribution of the two loss functions during patch optimization and is expected to impact the failure rate of VLAs in robotic task execution. In the main experiments, we set $\alpha_1$ to 0.8. We further evaluate EDPA with $\alpha_1$ set to 0, 0.2, 0.5, 0.8, and 1 to examine how different trade-offs between these objectives affect the performance of various VLAs in the LIBERO simulation benchmark.

The results are summarized in Figure 6. EDPA generates effective adversarial patches for OpenVLA across all $\alpha_1$ settings, indicating that both loss terms contribute to attacks that the model cannot resist. On OpenVLA-OFT, the highest failure rate occurs at $\alpha_1 = 0.2$, but variations in $\alpha_1$ do not result in substantial differences in attack performance. However, the failure rate of $\pi_0$ increases clearly with higher $\alpha_1$, suggesting that a larger contribution of $\mathcal{L}_{\text{patch}}$ enhances the effectiveness of EDPA on this model. In summary, the effectiveness of EDPA varies across different VLA models. The sensitivity of each model to the hyperparameter $\alpha_1$ differs, suggesting that the two loss functions have varying effectiveness depending on the target VLA.

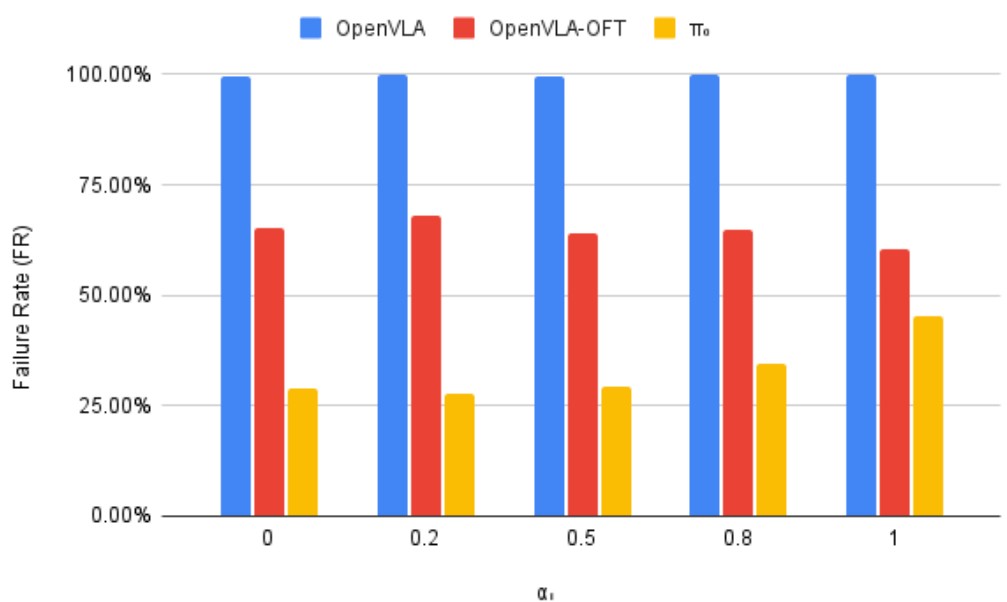

Figure 6: **Impact of hyperparameter** $\alpha_1$**.** The figure illustrates how varying $\alpha_1$ affects the performance of EDPA in terms of average failure rate (FR) across different VLA models on the LIBERO simulation benchmark.

## D LARGE LANGUANGE MODEL USAGE STATEMENT

The large language model was only used to polish the language during the preparation of this manuscript. Specifically, we used the well-known LLM ChatGPT[1] for this purpose. The model was not used to generate any technical content, ideas, or analyses.

---

[1]https://chatgpt.com

