# OpenReview forum: "Model-agnostic Adversarial Attack and Defense for Vision-Language-Action Models"
_ICLR.cc/2026/Conference — Submitted to ICLR 2026_

### Official Review · Reviewer_oTup · 2025-10-28

**Soundness:** 2
**Presentation:** 2
**Contribution:** 1
**Rating:** 2
**Confidence:** 4

**Summary:**

This paper investigates the robustness of Vision-Language-Action (VLA) models under adversarial attacks. The authors propose a method called Disruption Patch Attack (EDPA), which learns adversarial patches by enlarging the distance between adversarial and original images, as well as between adversarial images and their corresponding text descriptions. Experimental results show that the proposed method degrades VLA performance.

**Strengths:**

1. The paper focuses on an important and timely topic, i.e., evaluating the robustness of VLA models.
2. Experimental results demonstrate that the proposed method can successfully reduce the performance of VLA systems.

**Weaknesses:**

1. The novelty of the proposed method is limited. Learning adversarial examples by maximizing the distance from the original data is a common approach in adversarial attack literature [1-3].
2. Although the work targets VLA systems, it does not analyze how visual perturbations affect the action generation process. As a result, the proposed attack closely resembles conventional vision-language model attacks.
3. While the authors claim that their method does not require access to target models, it still relies on the use of encoders.
4. Experiments are conducted on a single benchmark with only a few baseline comparisons.

[1] Jiaming Zhang, Qi Yi, and Jitao Sang. Towards adversarial attack on vision-language pre-training models. In Proceedings of the 30th ACM International Conference on Multimedia, pp. 5005-5013, 2022.
[2] Zhou Z, Hu S, Li M, et al. Advclip: Downstream-agnostic adversarial examples in multimodal contrastive learning[C]//Proceedings of the 31st ACM International Conference on Multimedia. 2023: 6311-6320.
[3] Zhang, Peng-Fei, Zi Huang, and Guangdong Bai. "Universal adversarial perturbations for vision-language pre-trained models." Proceedings of the 47th International ACM SIGIR Conference on Research and Development in Information Retrieval. 2024.

**Questions:**

Please refer to Weaknesses.

---

> ### Author Response · Authors · 2025-11-21
>
> We would like to sincerely thank the reviewers for your time and insightful comments on our paper, and we provide responses to the weaknesses and questions raised below:
>
> &nbsp;
>
> **W1. Learning adversarial examples by maximizing the distance from the original data is a common approach in adversarial attack literature [1-3].**
>
> As you mentioned, increasing embedding distance is indeed a common strategy to attack vision-language model. However, these approaches [1,2,3] are targeting on contrastive learning models (e.g., CLIP), which do not use a language-model backbone. These models consist of modality-specific encoders that map each input to a single embedding vector, and the training objective is to bring paired embeddings closer in the embedding space. Accordingly, these methods design adversarial examples around the contrastive loss and evaluate on benchmarks tailored for contrastive models, such as image–text retrieval.
>
> In contrast, VLA models with a language-model backbone operate very differently. Inputs are not encoded into one embedding vector; they are tokenized, and the encoder produces token-level embeddings for each token. These token embeddings are then processed by the VLM backbone, whose generative objective is to predict robot actions that match labelled demonstrations. This mechanism differs fundamentally from contrastive models, which only optimize global embedding similarity.
>
> Therefore, our method is conceptually distinct: we aim to disrupt the model’s understanding of the input, rather than directly manipulating the final output. Our loss functions operate at the token level, modifying the distribution of intra modal and cross modal token similarities. By changing how the language model interprets relationships among tokens, we indirectly influence its final action predictions.
>
> &nbsp;
>
> **W2.  It does not analyze how visual perturbations affect the action generation process. As a result, the proposed attack closely resembles conventional vision-language model attacks.**
>
> In our response to W1, we explained the fundamental differences between our method and traditional attacks on vision-language models. Therefore, we argue that the claim that EDPA “closely resembles conventional vision-language model attacks” is not well-founded.
>
> In Appendix A, we visualize the average attention weights of the first and last layers of the VLA’s language-model backbone. We observe that EDPA-generated patches lead to clear shifts in attention distributions at both layers. Although these shifts are not the direct cause of the final action predictions, they show that EDPA alters internal token representations and attention patterns, which in turn indirectly influence final action generation.
>
> &nbsp;
>
> **W3. While the authors claim that their method does not require access to target models, it still relies on the use of encoders.**
>
> We would like to clarify that we did not claim our method requires no access to the target model, nor did we assume that an attacker could conduct EDPA in a fully black-box setting. Our point is that EDPA does not require the adversary to know the internal design of the victim model or the robotic platform it controls, which makes EDPA more practical and broadly applicable than prior methods (i.e. UADA and UPA).
>
> In general, understanding a model’s internal architecture is far more difficult than simply interacting with it, just as understanding the internals of a Transformer is much harder than using a packaged API.
>
> In parallel, UADA and UPA are tightly coupled to the OpenVLA architecture. These methods rely on the OpenVLA design, which abstracts continuous action predictions into a classification problem by discretizing action values into class labels and representing each label with a corresponding token, where the numerical differences between labels indicate the relative distances between actions in physical space. As a result, models that do not adopt this design, such as OpenVLA-OFT which no longer uses discrete action tokens, cannot support UADA and UPA.
>
> &nbsp;
>
> **W4. Experiments are conducted on a single benchmark with only a few baseline comparisons.**
>
> To the best of our knowledge, the set of reasonable baselines available for comparison in this field is quite limited. Most traditional adversarial attacks are not physically deployable, making them misaligned with the real-world challenges that VLA systems are designed to address. For this reason, both Wang et al. and our work do not adopt these methods as comparable baselines. We welcome suggestions for additional baselines that may be more closely aligned with our settings and will consider including them in future comparisons.
>
> Due to limitations in space, budget, and hardware availability, we are currently unable to conduct experiments on a real physical platform. We apologize for this constraint and therefore evaluate our method only in simulation benchmarks at this stage.

---

> > ### Comment · Reviewer_oTup · 2025-11-27
> > **Response to Authors**
> >
> > Thank you for the authors’ response. However, my concerns remain unaddressed. In particular, the authors do not clearly articulate the core differences between their method and existing conventional approaches; the proposed method still largely follows classical schemes. Given this, a more thorough comparison with established methods in the field is necessary. Therefore, I will keep my original score.

---

> ### Author Response · Authors · 2025-12-02
>
> Thank you very much for your feedback.
>
> Regarding your comment that “the proposed method still largely follows classical schemes,” our previous response has provided a detailed explanation of the key technical differences between our work and prior methods. If our earlier response did not fully address your concerns, we would greatly appreciate it if you could kindly indicate which technical aspects you feel are particularly close to existing work. This would help us better understand your perspective and further improve the manuscript..
>
> If there are any remaining concerns that we have not fully addressed, we would sincerely appreciate it if you could specify them. We are committed to doing our best to resolve all of your concerns.
>
> Thank you again for your time and constructive comments.

---

### Official Review · Reviewer_Who3 · 2025-10-31

**Soundness:** 3
**Presentation:** 3
**Contribution:** 2
**Rating:** 2
**Confidence:** 4

**Summary:**

This paper introduces a model-agnostic adversarial attack against vision-language-action (VLA) models that is designed to elicit task failure. By using an attack that only targets the image and language encoders, this attack requires less information than previous state-of-the-art attacks designed to elicit the same objective. Finally, the authors present a method to fine-tune the visual encoder to improve robustness.

**Strengths:**

This paper is technically sound and well presented. The proposed attack does improve the feasibility and ease of optimization compared to previous methods, and achieves better ASR than previous work. Additionally, this is the first (to my knowledge) work that introduces a feasible defense to the proposed adversarial attack. Such defenses are incredibly useful to the field, given the growing number of papers evaluating adversarial attacks against VLAs.

**Weaknesses:**

While this attack claims to require less information than prior attacks to succeed, namely access to the encoders only, this remains a white-box attack. In table 1, the authors compare with attacks from [Wang et al.](https://arxiv.org/pdf/2411.13587) which require knowledge of the VLA's action space, robotic manipulator, and access to the encoder parameters, and LVLM parameters. However, unfortunately they do not sufficiently motivate a threat model where an adversary might have access to the encoder parameters and not the LVLM parameters, or knowledge of the other two requirements for UADA and UPA. Therefore, the main contributions of this paper are state of the art results in terms of ASR on the LIBERO datasets, and a feasible defense mechanism for this attack. However, UPA, the previous SOTA from Wang et al., already achieved near-perfect ASR on the "task failure" objective, making the small percentage increase not a sufficiently interesting result.

Additionally, given the fact that none of the evaluated models can reliably complete every task in the LIBERO dataset, "task failure" is not a meaningful metric for evaluation. Further, unlike Wang et al., the authors do not present real-world results, which significantly weakens their claims.

Finally, while their demonstration of adversarial fine-tuning is undoubtedly a useful addition to the paper, it is a technique that is already well represented in adversarial machine learning literature.

**Questions:**

1. Why do you think the proposed defense performs so poorly against UADA?
2. Adversarial fine-tuning works better on some datasets (Spatial, Object) compared to others (Goal, Long). Do you have any intuition for why this might be the case?

---

> ### Author Response · Authors · 2025-11-21
>
> We would like to sincerely thank the reviewers for your time and insightful comments on our paper, and we provide responses to the weaknesses and questions raised below:
>
> &nbsp;
>
> **W1. Do not sufficiently motivate a threat model where an adversary might have access to the encoder parameters and not the LVLM parameters, or knowledge of the other two requirements for UADA and UPA**
>
> We are not focus white/black-box settings, but rather on the practicality and general applicability of attack methods. In this context, we believe that understanding a model’s internal design is substantially more difficult than directly using the model itself, much like how understanding the internals of a Transformer is far more challenging than simply calling a packaged API.
>
> In parallel, UADA and UPA are specifically designed for the OpenVLA architecture. These methods rely on the OpenVLA design, which abstracts continuous action predictions into a classification problem by discretizing action values into class labels and representing each label with a corresponding token, where the numerical differences between labels indicate the relative distances between actions in physical space. As a result, models that do not adopt this design, such as OpenVLA-OFT which no longer uses discrete action tokens, cannot support UADA and UPA.
>
> &nbsp;
>
> **W2. The previous SOTA from Wang et al., already achieved near-perfect ASR on the "task failure" objective, making the small percentage increase not a sufficiently interesting result.**
>
> We believe that improving performance by only percentage points over existing SOTA methods should not be the sole criterion for evaluating a contribution. As discussed in our response to W1, the key value of EDPA lies in its generality and practicality, rather than small metric gains over UADA or UPA.
>
> In Table 1, our goal is to show that even though UADA and UPA are specifically designed for OpenVLA, EDPA still achieves near-perfect attack performance comparable to them.
>
> &nbsp;
>
> **W3. Given the fact that none of the evaluated models can reliably complete every task in the LIBERO dataset, "task failure" is not a meaningful metric for evaluation.**
>
> Both Wang et al. and we chose to use the failure rate because, indeed, no evaluated model can successfully complete every task. Comparing failure rates under clean and perturbed conditions allows us to objectively quantify how much performance degrades when the VLA models are exposed to adversarial patches.
>
> &nbsp;
>
> **W4. Unlike Wang et al., the authors do not present real-world results, which significantly weakens their claims.**
>
> Due to limitations in space, budget, and hardware availability, we are currently unable to conduct experiments on a real physical platform. We apologize for this constraint and therefore evaluate our method only in simulation benchmarks at this stage.
>
> &nbsp;
>
> **W5. Adversarial fine-tuning is undoubtedly a useful addition to the paper, it is a technique that is already well represented in adversarial machine learning literature.**
>
> While adversarial training is well-studied, most existing methods focus on models and problems different from ours. Our adversarial fine-tuning differs from traditional training in the following ways:
>
> 1.	Traditional adversarial training generates adversarial examples using pixel-level attacks (e.g., PGD) to improve robustness against pixel perturbations. In contrast, we construct adversarial examples using EDPA-generated patches. Due to the substantial difference between pixel-level and patch-based attacks, our approach involves several technical modifications not present in conventional adversarial training.
>
> 2.	Mainstream adversarial training typically updates the entire model, whereas we fine-tune only the visual encoder to enhance overall model robustness. Accordingly, the learning objective we use differs significantly from standard adversarial training.
>
> The technical details of our method are presented in Section 3.3, where reviewers can find a comprehensive technical description of our approach.
>
> &nbsp;
>
> **Questions: Why defense performs so poorly against UADA and Adversarial fine-tuning works better on some datasets (Spatial, Object) compared to others (Goal, Long).**
>
> We observed visual encoders finetuning on the Goal and Long tasks experienced extreme loss spikes, which prevented recovery to pre-spike levels and led to weaker defense performance compared to the Spatial and Object tasks. To address this, we adopted a simpler and stable method: rolling back to the checkpoint before the spike.  We are currently redoing the fine-tuning of the Goal and Long models. Retraining on Goal has completed, and we observe a significant improvement in its robustness, including against UADA and UPA.
>
> We plan to update these preliminary evaluation results in the rebuttal within the next two days and will provide the full results from three repeated experiments in the next version of the paper.

---

### Official Review · Reviewer_7HPh · 2025-11-01

**Soundness:** 3
**Presentation:** 3
**Contribution:** 3
**Rating:** 6
**Confidence:** 4

**Summary:**

This paper proposes a model-agnostic adversarial patch attack method EDPA, requiring only access to the VLA's visual encoder parameters. EDPA generates a patch by  maximizing the discrepancy between the embeddings of clean and perturbed visual inputs and disrupting the semantic alignment between the visual embeddings and the language instruction's textual embeddings. The paper also provides an adversarial fine-tuning scheme for the visual encoder for defense, in which the encoder is optimized to generate similar latent representations for both clean and adversarially perturbed visual inputs. The results show that EDPA decreases the task success rate of VLA models, while the proposed defense mitigates this degradation.

**Strengths:**

1. The design of the EDPA loss function is well-conceived. Targeting the semantic alignment between visual and textual latent representations is a logical and novel way to attack the core mechanism of a VLA, rather than just its final action outputs.

2. The experiments are comprehensive, which evaluate against multiple, relevant SOTA models under different settings.

**Weaknesses:**

1. Limited Scope of Defense Evaluation: The proposed adversarial fine-tuning defense is only evaluated on the single-camera OpenVLA model. While the paper attacks multi-camera models (OpenVLA-OFT and $\pi_{0}$), it does not demonstrate whether the defense is effective for them. These models are more complex and, as the paper notes, already exhibit higher baseline robustness. It is unclear how the defense would be applied or how effective it would be.

2. Lack of Defense Baselines: The paper proposes an adversarial fine-tuning scheme but does not compare it against other defense methods.

3. Patch Occlusion Problem: The authors identify in their limitations that the patch "may occasionally occlude important objects". For example, this could cause the adversarial fine-tuning to train the model to ignore an object that the patch is covering. This is a non-trivial problem for the defense.

**Questions:**

1. Your defense evaluation was focused on the single-camera OpenVLA model. How would you adapt your adversarial fine-tuning scheme for a multi-camera model like $\pi_{0}$
​
2. You propose a hypothesis that VLA encoders overfit to the robotic arm's appearance. Have you considered an experiment to directly test this? For example, one could fine-tune a VLA model on a dataset where the robotic arm is masked out or digitally removed. If your hypothesis is correct, this model should be more robust to your patch attack, even without adversarial fine-tuning.

3. You note as a limitation that the adversarial patch may occlude important objects. In your current experiments, how is the patch placed? Is it at a fixed location (as in the Appendix attention visualizations ) or placed randomly? Could the EDPA attack be made stronger by jointly optimizing for patch location to maximize disruption (e.g., placing it near the arm) while avoiding occlusion of the target object mentioned in the instruction?

---

> ### Author Response · Authors · 2025-11-21
>
> We would like to sincerely thank the reviewers for your time and insightful comments on our paper, and we provide responses to the weaknesses and questions raised below:
>
> &nbsp;
>
>
> **W1. The proposed adversarial fine-tuning defense is only evaluated on the single-camera OpenVLA model. While the paper attacks multi-camera models (OpenVLA-OFT and Pi0), it does not demonstrate whether the defense is effective for them.**
>
> We are currently conducting adversarial fine-tuning on both OpenVLA-OFT and pi0. However, fine-tuning requires significant computational resources and detailed familiarity with each model’s codebase, as separate scripts must be implemented for each model. As a result, these experiments need additional time.
>
> Currently, fine-tuning on OpenVLA-OFT has been completed, and the final evaluation is underway, with preliminary results showing promising performance. We plan to update these initial re-evaluation results in the rebuttal within the next two days and provide the full results from three repeated experiments in the next version of the paper.
>
> &nbsp;
>
> **W2. Lack of Defense Baselines**
>
> Given the limited work in this area, we have not identified suitable defense baselines for comparison. Wang et al. evaluated some more traditional defense methods, but these were largely ineffective against such attacks. If time allows, we plan to supplement our experiments with these defense methods to provide a more complete comparison. Also, we welcome suggestions for additional baselines that may be more closely aligned with our settings and will consider including them in future comparisons.
>
> &nbsp;
>
> **W3. Patch Occlusion Problem**
>
> We acknowledge that this is indeed a potential risk, especially when the chosen patch size increases, which also raises the likelihood of such issues during fine-tuning.
> However, based on our current experimental results, adversarial fine-tuning remains highly robust. This is likely because the patches used during fine-tuning are only 50×50 pixels (approximately 4% of the entire image) and are placed randomly within the image. As a result, only a very small fraction of adversarial samples fully occlude the important object during fine-tuning, and overall performance is hardly affected.
>
> &nbsp;
>
> **Q1 . How would you adapt your adversarial fine-tuning scheme for a multi-camera model like pi0.**
>
> These multi-camera models, including OpenVLA-OFT and pi0, use the same visual encoder to process images from different viewpoints. Our fine-tuning approach is straightforward: we include images from all viewpoints as training data during adversarial fine-tuning.
>
> &nbsp;
>
> **Q2. You propose a hypothesis that VLA encoders overfit to the robotic arm's appearance. Have you considered an experiment to directly test this? For example, one could fine-tune a VLA model on a dataset where the robotic arm is masked out or digitally removed.**
>
> We conducted a preliminary experiment: the base models of OpenVLA and pi0 were both obtained by fine-tuning a pre-trained VLM on large-scale robotic data. Using the visual encoders of these models, we generated patches via the EDPA method, which resulted in chaotic patches containing almost no semantic information.
>
> We did consider directly fine-tuning the VLA models, but ultimately did not adopt this approach. The main reason is that significantly altering the camera viewpoints or digitally removing the robotic arm during fine-tuning would create a data distribution that differs substantially from the one used to train the base model, potentially degrading the model’s performance on tasks it could previously perform successfully. Additionally, fine-tuning the entire VLA model (without using LoRA) is extremely resource-intensive, making such experiments practically infeasible.
>
> &nbsp;
>
> **Q3. You note as a limitation that the adversarial patch may occlude important objects. In your current experiments, how is the patch placed?**
>
> To ensure that our patch is effective at any location, the position of the patch is randomized during both patch generation and evaluation (line 193). In Appendix A, we placed the patch in the top-left corner solely to better visualize changes in attention.

---

### Official Review · Reviewer_74Pq · 2025-11-05

**Soundness:** 2
**Presentation:** 2
**Contribution:** 1
**Rating:** 2
**Confidence:** 4

**Summary:**

This paper proposes an adversarial patch attack for VLAs. The authors describe methods for crafting adversarial patches, and for fine-tuning models to be resistant against these patch attacks. Experiments are presented on LIBERO for a handful of models showing the effectiveness of the attack.

**Strengths:**

- I agree with the authors that this problem is understudied, and that patch attacks are a reasonable place to start.

**Weaknesses:**

- The related work on adversarial attacks on VLAs seems a bit sparse. There have been several works beyond (Wang et al., 2024) that tackle this problem; see, e.g., https://arxiv.org/pdf/2505.16640? and https://arxiv.org/abs/2506.03350. Additionally, while the authors cite related work from the adversarial examples community, there is work much closer to the robotics community on attacking LLM-based planners, see, e.g., https://arxiv.org/abs/2407.20242 and https://arxiv.org/abs/2410.13691. I think adding a discussion of these works, particularly those on VLA attacks, would give a better view of the literature, particularly because as written, one could reasonably infer that the only work on VLA attacks is Wang et al., 2024.
- I don't understand why the baselines differ between Table 2 and Table 3. The paper feels incomplete by selectively including these baselines. Furthermore, why did the authors not consider fine-tuning OpenVLA-OFT and pi0 to improve robustness?
- For a purely empirical paper, this paper is *very* light on results, so much so that I'm going to advocate for rejection. It is reasonable to expect that an accepted paper should offer more creative/fine-grained analysis. The paper, as written, doesn't make a compelling/strong case that this method is significantly stronger than the baselines due to a lack of evidence.
- The results also don't seem particularly compelling. It's somewhat hard to interpret the results in Table 3, because the authors use weak baselines (clean performance and random patches). And it looks like the baselines in Table 2 from Wang et al., 2024 do essentially the same. Perhaps there's an argument to be made that the patch attack uses a slightly more realistic threat model, but I think even this is arguable, given that it seems unlikely that a real adversary will apply patches in practice.
- Since we're talking about robotics, why not try this on real hardware? That would certainly result in further impact than showing that this works in simulation. If I remember correctly, the motivation behind patch attacks back in the 2010s was that someone could insert/place patches onto stop signs or t-shirts, causing AVs to perform unsafe actions (happy to provide references if helpful).

**Questions:**

See above.

---

> ### Author Response · Authors · 2025-11-21
>
> We would like to sincerely thank the reviewers for your time and insightful comments on our paper, and we provide responses to the weaknesses and questions raised below:
>
> &nbsp;
>
>
> **W1. The related work on adversarial attacks on VLAs seems a bit sparse.**
>
> We first thank the reviewer for providing many valuable references. We would like to clarify that these works were not included in our related work section because they do not fully align with the scope of our study, which focuses on physically realizable adversarial attacks on VLA models.
>
> For example, [1] studies backdoor attacks, which are not within the scope of adversarial attacks; their approach injects triggers into the training set to make the VLA learn undesirable behaviors. The works [2, 3, 4] are jailbreak attacks, which are achieved by modifying user instructions or prompts. These attacks are neither strictly defined adversarial attacks nor physically realizable, similar to pixel-level attacks.
>
> To conserve space and maintain focus, the main text includes only work directly relevant to our settings. A more detailed discussion of these VLA-related challenges will be provided in the appendix in the next version.
>
> &nbsp;
>
> **W2. Why the baselines differ between Table 2 and Table 3. The paper feels incomplete by selectively including these baselines.**
>
> The baselines in Table 2 and Table 3 differ because UADA and UPA are specifically designed for the OpenVLA architecture and are difficult or impossible to transfer to other VLA models. This is because these methods rely on the OpenVLA design, which abstracts continuous action predictions into a classification problem by discretizing action values into class labels and representing each label with a corresponding token, where the numerical differences between labels indicate the relative distances between actions in physical space. As a result, models that do not adopt this design, such as OpenVLA-OFT which no longer uses discrete action tokens, cannot support UADA and UPA, explaining their absence in Table 3.
>
> &nbsp;
>
> **W3. Why not consider fine-tuning OpenVLA-OFT and pi0 to improve robustness?**
>
> We are currently conducting adversarial fine-tuning on both OpenVLA-OFT and pi0. However, fine-tuning requires significant computational resources and detailed familiarity with each model’s codebase, as separate scripts must be implemented for each model. As a result, these experiments need additional time.
> Fine-tuning on OpenVLA-OFT is already completed, and the final evaluation is in progress, with preliminary results showing strong performance. We plan to include these initial results in the rebuttal within the next two days, and provide full results from three runs in the next paper revision.
>
> &nbsp;
>
> **W4. The results also don't seem particularly compelling. It's somewhat hard to interpret the results in Table 3, because the authors use weak baselines.**
>
> To the best of our knowledge, the set of reasonable baselines available for comparison in this field is quite limited. Most traditional adversarial attacks are not physically deployable, making them misaligned with the real-world challenges that VLA systems are designed to address. For this reason, both Wang et al. and our work do not adopt these methods as comparable baselines. We welcome suggestions for additional baselines that may be more closely aligned with our settings and will consider including them in future comparisons.
>
> &nbsp;
>
> **W5. Why not try this on real hardware?**
>
> Yes. Our motivation for choosing a patch-based attack is precisely its physical realizability. Since VLA models are designed to operate in the physical world, we believe patch-based attacks more accurately reflect the types of challenges VLA systems are likely to face in practice.
> However, due to limitations in space, budget, and hardware availability, we are currently unable to conduct experiments on a real physical platform. We apologize for this constraint and therefore evaluate our method only in simulation benchmarks at this stage.
>
> &nbsp;
>
> [1] Zhou, X., Tie, G., Zhang, G., Wang, H., Zhou, P., & Sun, L. (2025). BadVLA: Towards Backdoor Attacks on Vision-Language-Action Models via Objective-Decoupled Optimization. arXiv preprint arXiv:2505.16640.
>
> [2] Jones, E. K., Robey, A., Zou, A., Ravichandran, Z., Pappas, G. J., Hassani, H., ... & Kolter, J. Z. (2025). Adversarial Attacks on Robotic Vision Language Action Models. arXiv preprint arXiv:2506.03350.
>
> [3] Zhang, H., Zhu, C., Wang, X., Zhou, Z., Yin, C., Li, M., ... & Zhang, L. Y. (2024). BadRobot: Jailbreaking embodied LLMs in the physical world. arXiv preprint arXiv:2407.20242.
>
> [4] Robey, A., Ravichandran, Z., Kumar, V., Hassani, H., & Pappas, G. J. (2025, May). Jailbreaking llm-controlled robots. In 2025 IEEE International Conference on Robotics and Automation (ICRA) (pp. 11948-11956). IEEE.

---

> > ### Comment · Reviewer_74Pq · 2025-11-25
> > **A response to your rebuttal**
> >
> > Thanks so much for your rebuttal. Here are a few comments on my end.
> >
> > * **W1.** I'm a bit confused. The authors claim that "backdoor attacks are not within the scope of adversarial attacks." This is *not* generally correct in the context of ML security. While backdoor attacks do employ a different threat model than input-based attacks, they are still, by their nature, adversarial. Furthermore, I'd like to gently push back on the claim that "jailbreaking attacks. . . are neither strictly defined adversarial attacks nor physically realizable." Some of the references in your response run the attacks on physical hardware: [3] runs on a UR3e manipulator and [4] runs on the Unitree Go2 robot. This indicates that the attacks *are* physically realizable.
> >
> > * **W2.** Ok, that makes sense. It would be worth clearly articulating this in the paper.
> >
> > * **W3.** Any update on the results?
> >
> > * **W4 and W5.** See the comments in response to **W1** re:physically realizable. To expand on this: If the claim is that other attacks aren't physically realizable, it seems reasonable to expect evidence that the attack proposed in this paper *is* physically realizable.
> >
> > Given the current state of the paper, I will maintain my score.

---

> > > ### Author Response · Authors · 2025-12-03
> > >
> > > Thank you very much for your feedback.
> > >
> > > We acknowledge the broader taxonomy of adversarial attack in machine learning. However, in the many ML security literature (e.g. [5]), the term “adversarial attack” is typically used to refer to adversarial example attacks, i.e., inference-time input perturbations. In contrast, backdoor and data-poisoning attacks rely on fundamentally different training-time assumptions and are therefore categorized as separate security threats.
> > >
> > > We believe that prompt-based attacks [2,3,4] are not physically realizable primarily because they typically assume that the attacker can directly modify the user’s input (e.g., textual commands or prompts), like pixel-level attack we mentioned in the paper. This assumption is difficult to satisfy in real world scenarios, as attackers generally do not have the access to directly alter user inputs. Although these works include certain physical experiments, their attack strategies fundamentally rely on direct manipulation of user inputs,
> > >
> > > We are currently unable to conduct experiments in the physical world. However,  the Wang et al. already demonstrated successful physical experiments, it provides strong evidence that the patch attack can be physically realizable.
> > >
> > > Regarding the additional defense experiments you mentioned, we provide the fine-tuning results for OpenVLA-OFT on the top of this page.
> > >
> > > Reference
> > > [5] Xue, M., Yuan, C., Wu, H., Zhang, Y., & Liu, W. (2020). Machine learning security: Threats, countermeasures, and evaluations. IEEE Access, 8, 74720-74742.

---

### Author Response · Authors · 2025-12-03

We would like to thank all reviewers for their valuable comments. Below, we provide the additional experimental results requested by the reviewers.

**1. Results of adversarial finetuning on LIBERO-Goal and LIBERO-Long**

In the main submission, the finetuned OpenVLA did not show strong improvements on LIBERO-Goal and LIBERO-Long. Upon further investigation, we found that the performance instability was caused by loss spikes during finetuning, which significantly impacted the results. To address this issue, we applied a rollback strategy: whenever an abnormal surge in loss was detected, we reverted the model parameters to a previously stable checkpoint and resumed training. This stabilized the training and led to substantial improvements on both task suites.

The table below reports the failure rates after applying adversarial finetuning:

| Source | Clean | Random | UADA | UPA | EDPA |
|--------|-------|--------|------|-----|------|
| Goal   | 22.6 (**-4.3**) | 30.8 (**-7.1**) | 26.8 (**-71.8**) | 29.4 (**-69.5**) | 32.0 (**-68.0**) |
| Long   | 49.0 (**+0.9**) | 49.0 (**-25.9**) | 50.0 (**-49.6**) | 51.8 (**-47.8**) | 61.6 (**-38.4**) |

The values in parentheses indicate the change compared to the non-finetuned results.

**2. Results of adversarial finetuning on OpenVLA-OFT across all LIBERO simulation task suites**

We also applied adversarial finetuning to the OpenVLA-OFT model and evaluated it on all four LIBERO simulation task suites. The table below reports the failure rates after finetuning:

| Source  | Clean        | Random       | EDPA         |
|---------|-------------|-------------|-------------|
| Spatial | 2.2 (**+0.8**)  | 2.4 (**−5.7**)  | 2.8 (**−36.9**) |
| Object  | 2.0 (**0.0**)   | 2.0 (**−13.9**) | 4.0 (**−48.3**) |
| Goal    | 2.8 (**0.0**)   | 3.6 (**−1.5**)  | 13.2 (**−67.6**) |
| Long    | 5.4 (**+0.5**)  | 7.0 (**−21.1**) | 9.4 (**−77.0**) |

The values in parentheses indicate the change compared to the non-finetuned results.

Due to computational and time constraints, we have not yet completed adversarial finetuning experiments for pi0. However, we believe that the existing results already sufficiently demonstrate the effectiveness of adversarial finetuning in improving model robustness. Once again, we sincerely thank the reviewers for their valuable suggestions!

---

### Meta-Review · Area_Chair_8kvC · 2026-01-07

**Summary:**

Reviewers agree the paper tackles an important problem in VLA robustness and that the proposed EDPA attack and adversarial fine-tuning defense are technically sound and effective in simulation, but most reviewers remain unconvinced by the overall strength of the contribution due to limited novelty over existing adversarial attack methods, reliance on a single benchmark, lack of real-world validation, and insufficiently compelling empirical evidence.

**Reviewer Concerns:**

The rebuttal and added experiments strengthen the defense evaluation, especially for OpenVLA-OFT, and clarify baseline differences, but core concerns remain largely unresolved, including sparse and disputed related work positioning, debatable threat model assumptions, limited novelty relative to classical attacks, weak or incomplete baselines, and the absence of physical-world experiments despite claims of physical realizability.

**Reviewer Scores:**

Reviewer 7HPh (only positive) would likely remain marginally positive given the additional defense results, while Reviewers 74Pq and oTup (three negatives) explicitly indicate that their reject recommendations would not change, as concerns about novelty, evaluation strength, and physical validation persist after the rebuttal.

---

### Decision · Program_Chairs · 2026-01-26

Reject